# Homo-Acetogens: Their Metabolism and Competitive Relationship with Hydrogenotrophic Methanogens

**DOI:** 10.3390/microorganisms10020397

**Published:** 2022-02-08

**Authors:** Supriya Karekar, Renan Stefanini, Birgitte Ahring

**Affiliations:** 1Bioproducts Science and Engineering Laboratory, Washington State University Tri-Cities, 2720 Crimson Way, Richland, WA 99354, USA; s.karekar@wsu.edu (S.K.); r.stefaninilopes@wsu.edu (R.S.); 2Department of Biological Systems Engineering, Washington State University, Pullman, WA 99163, USA; 3The Voiland School of Chemical Engineering and Bioengineering, Washington State University, Pullman, WA 99163, USA

**Keywords:** acetogens, carbon dioxide, hydrogen, methane emissions, methanogens, ruminants

## Abstract

Homo-acetogens are microbes that have the ability to grow on gaseous substrates such as H_2_/CO_2_/CO and produce acetic acid as the main product of their metabolism through a metabolic process called reductive acetogenesis. These acetogens are dispersed in nature and are found to grow in various biotopes on land, water and sediments. They are also commonly found in the gastro-intestinal track of herbivores that rely on a symbiotic relationship with microbes in order to breakdown lignocellulosic biomass to provide the animal with nutrients and energy. For this motive, the fermentation scheme that occurs in the rumen has been described equivalent to a consolidated bioprocessing fermentation for the production of bioproducts derived from livestock. This paper reviews current knowledge of homo-acetogenesis and its potential to improve efficiency in the rumen for production of bioproducts by replacing methanogens, the principal H_2_-scavengers in the rumen, thus serving as a form of carbon sink by deviating the formation of methane into bioproducts. In this review, we discuss the main strategies employed by the livestock industry to achieve methanogenesis inhibition, and also explore homo-acetogenic microorganisms and evaluate the members for potential traits and characteristics that may favor competitive advantage over methanogenesis, making them prospective candidates for competing with methanogens in ruminant animals.

## 1. Introduction to Homo-Acetogens

Homo-acetogens are bacteria known for being capable to have a versatile metabolism for diverse substrate consumption, including the ability to grow on gaseous substrates using the acetyl-CoA pathway for fixing carbon dioxide (CO_2_) using hydrogen (H_2_) for synthesizing acetyl-CoA. It is this pathway that defines an homo-acetogen and is called the Wood–Ljungdahl (WD) pathway, which has been studied over the years, especially for the direct fixation of CO_2_ by the acetyl-CoA synthase enzyme to produce acetate as the energy molecule [1]. Acetyl-CoA synthase enzyme is the principal enzyme that enables fixation of gaseous substrates—H_2_/CO_2_/CO (carbon monoxide)—to mainly produce acetic acid. Along with acetic acid, other products such as ethanol, butyric acid, and butanol can also be produced [2,3,4,5,6]. The acetyl-CoA synthase enzyme is not exclusive to acetogens, it is also found in other processes, including in sulfate reducers, methanogens, and hydrogenogens [7,8,9,10,11]. Homo-acetogens are mainly divided into 22 different genera, amongst which the genera of *Acetobacterium*, *Clostridium*, *Morella*, *Eubacterium*, *Sporomusa*, and *Ruminococcus* will be the focus of this paper. Homo-acetogens have been found to grow in diverse habitats with various other microbes including methanogens. This is because of their ability to grow autotrophically on H_2_/CO_2_/CO, and heterotrophically on a variety of substrates including hexoses, pentoses, alcohols, acids—formic acid, and methyl groups [12]. They are present in numerous anaerobic environments including sediments, soils, and gastrointestinal tracts of animals [13].

The acetogenic pathway was discovered by Wood and Ljungdahl and hence called the Wood–Ljungdahl pathway (Figure 1). The process of acetogenesis is comprised of two branches, namely: the eastern and the western branch, with both utilizing acetyl-CoA synthase to form acetate as the end product. In the western branch, one molecule of CO_2_ is reduced by one molecule of H_2_ to produce formate, which then follows a series of enzymatic reactions involving the uptake of two more molecules of H_2_ to yield methyl-CoFeSP. In the eastern branch, another molecule of CO_2_ is reduced to CO using one molecule of H_2_ by acetyl-CoA synthase enzyme. This enzyme is central to WL pathway since it catalyzes the conversion of both the products of the branches i.e., methyl-CoFeSP and CO to acetyl-CoA, which later is converted to acetic acid. Overall, four molecules of H_2_ and two molecules of CO_2_ are required for the completion of the hydrogenotrophic process of acetogenesis with a resultant Gibbs free energy (ΔG_0_′) of −105 kJ/moles [12].

## 2. Homo-Acetogens Niche

Homo-acetogens have been found to be active in an array of habitats including, sediments, rumen, soil, salt marsh, sludge, and intestinal tracts of animals (Figure 2). These bacteria have been further classified based on morphology, nutritional requirements, substrates, oxygen tolerance, metabolic pathways, physiological properties, and the genetics of the homo-acetogens. Globally, around 1013 kg of acetate is produced in anaerobic habitats of which 10% is produced autotrophically through the WL pathway [14].

The first instance of acetate formation from H_2_-CO_2_ in sewage sludge was unraveled by Fischer et al. in 1932 [15]. In 1936, Wieringa isolated the first acetogen named *Clostridium aceticum* (*C. aceticum*), a mesophilic, spore-forming bacterium from ditch mud [16,17,18]. In 1942, a second thermophilic, spore-forming, sugar-utilizing acetogen named *Clostridium thermoaceticum* (*C. thermoaceticum*) was isolated from horse manure. Later, Wood and Ljungdahl discovered the potential of *C. thermoaceticum* to produce acetate using H_2_-CO_2_ as substrate and was later renamed to *Morella thermoacetica* (*M. thermoacetica*) [19]. *Clostridium scatologenes* (*C. scatologenes*) was later discovered by Küsel et al. in 2000, which was an acetogen, already found in 1927 but not previously characterized for its ability to use H_2_-CO_2_ [20,21]. Eventual investigations on the product formation by acetyl-CoA pathway, which primarily defines an homo-acetogen, led to the discovery of 22 different bacterial genera and over 100 species of homo-acetogens [12]. Different types of homo-acetogens explored in this paper have been listed in Table 1.

### 2.1. Sedimentary Environments

Homo-acetogens can be encountered in a variety of sedimentary environments, including tundra wetlands and lake deposits [46,47,48,49]. In these sedimentary environments, anaerobic conditions prevail due to lack of oxygen exposure. The homo-acetogenic and other microbial populations are largely determined by abiotic conditions of these environments, such as the location, depth, acidity, temperature, and sediment compositions [50,51]. Anoxic conditions favor the generation of a number of electron donors by virtue of hydrolysis and fermentation reactions [52]. These electron donors facilitate the growth of homo-acetogens, methanogens, as well as other bacteria and archaea in sediments [50,53,54,55]. As the conditions change significantly in different sediments, they are one of the best-suited habitats for potential screening of new homo-acetogens with advantageous traits to compete with hydrogenotrophic methanogens.

When a variety of sediments from freshwater, lake, marine environments along with ditch mud, coal mine, oil mills, and plant roots were studied, a diverse population of homo-acetogens was identified. The most common homo-acetogens found in these habitats belonged to *Acetobacterium*, *Clostridium*, and *Sporomusa* genera. Homo-acetogens like *Acetobacterium woodii* (*A. woodii*), *Acetobacterium carbinolicum* (*A. carbinolicum*), and *Acetobacterium malicum* (*A.malicum*), were a few non-spore forming, Gram-negative acetogens found in the fresh water and marine sediments. *Clostridium* ssp. included, *Clostridium drakei* (*C. drakei*), *Clostridium glycolicum* (*C. glycolicum*), *Clostridium carboxidivorans* P7T (*C. carboxidivorans*), and *Clostridium scatologenes* (*C. scatologenes)*. Many of the homo-acetogens belonging to *Sporomusa* genus were also encountered which included the strains of *Sporomusa malonica* (*S. malonica*), *Sporomusa paucivorans* (*S. paucivorans*), *Sporomusa* sp. DR6, and *Sporomusa* sp. DR1/8, etc. [12]. All the homo-acetogens studied showed the ability to grow on H_2_-CO_2_; however, even though they performed well in laboratory settings, they did not seem to rely on this reaction in nature for their growth. This was due to the demand for higher concentration of H_2_ to utilize CO_2_, which was unavailable in these habitats. The H_2_ concentration in these anoxic habitats was in the range of 0.01–0.04 mbar [35,56,57,58,59] and the minimum threshold required by the homo-acetogens is around 0.43–0.95 mbar [60]. However, only under unfavorable conditions such as low temperatures, some of these homo-acetogens outgrew methanogens, e.g., *Acetobacterium* strain HP4. As the temperature was increased above 17 °C, the methanogenic population became dominant [56].

The reasons behind the existence of these abovementioned homo-acetogens in these environments may be attributed to—(1) the absence of methanogens competing for H_2_ and homo-acetogens grew very sparsely on whatever H_2_ was available or (2) their ability to grow heterotrophically on other nongaseous substrates. An exception to this pattern was observed in the case of increased acetogenic activity detected in peatlands. A study conducted by Ye et al. in 2014 showed that homo-acetogenesis was dominant in peatlands with 25–63% acetate production in fen peat followed by bog peat and swamp peat. Hydrogenotrophic methanogenesis also coexisted and became competitive in the later stages. However, moderate temperatures around 17 °C favored acetogenesis. Interspecies transfer of H_2_ was attributed as the main reason for increased acetogenesis in comparison to H_2_ transfer from the pore water in the case of methanogenesis [61].

### 2.2. Gastro-Intestinal Tract of Animals

The gastrointestinal (GI) tract of animals provides an ideal ecosystem for anaerobic microorganisms to thrive. The presence of this microbiota in the GI tract helps the host animal with nutrient absorbance from the ingested food, produces chemicals used for energy metabolism, and protects the host cells against pathogens [62,63].

Due to the recalcitrant nature of lignocellulosic biomass, many herbivores have developed digestion systems based on fermentation with symbionts, coupled with the capability to utilize their fermentative products for nutrition. These symbiotic digestion systems can be classified according to where the main fermentation occurs in the GI track, known as hindgut and foregut systems [64]. In hindgut fermenters, the breakdown of lignocellulosic biomass by microbes occurs in the large intestines, after it had passed through gastric digestion. Foregut fermenters, on the other hand, have an enlarged chamber adapted to accommodate fermenting microbes for the breakdown of biomass before gastric digestion [65].

#### 2.2.1. Hindgut Fermenters

In hindgut fermenters, the large intestines, especially the cecum, are usually larger in order to accommodate a high-density microbiota that is responsible for fermenting most nutrients that were not absorbed by the host in the upper intestinal tract [62,66].

This type of fermenter shows a great variation in the structure, size, and function of dedicated fermentation chambers, even among mammals. Following are some examples of hindgut fermenters that have prominent homo-acetogenic activity.

Homo-acetogens in Rabbits

Homo-acetogenic bacteria associated with reduced methane emissions were observed in rabbits in the cecum. When the H_2_ conversion to methane was compared between rabbit cecum and goat rumen, the rabbit cecum showed only 24% H_2_ conversion into methane compared to 85% in goat rumen. Therefore, the ceca of the rabbits were examined for the presence of novel homo-acetogens. Young and adult rabbits in four different age groups were studied. It was observed that the homo-acetogenic population in adult rabbits was more diverse in comparison to young rabbits. However, homo-acetogens were found to be more constant in adult rabbits, which could be attributed to stable microbial environments with age [3,67,68].

When the expression of mcrA gene for methanogens and fhs gene for homo-acetogens were compared, it showed prevalence of the gene copy number of 10^6^–10^8^ of mcrA gene copy numbers per gram of cecal content and 10^5^–10^6^ of fhs gene copy numbers per gram of cecal content, thereby indicating a higher occurrence of methanogens compared to homo-acetogens [3]. The methanogenic population was found to increase with the age of the rabbit; however, the acetogenic population remained more or less steady for all age groups [27,69]. The community of methanogens was particularly dominated by one single genus, *Methanobrevibacter*. Notable versatility in the population of homo-acetogens was observed in the cecum of rabbits which was mainly inhabited by Lachnospiraceae, Ruminococcaceae, and Clostridiaceae. The *Blautia* genus was found to be dominant, including *Blautia coccoides (B. coccoides)*, *Blautia schinkii* (*B. schinkii*), *Blautia hydrogenotrophica* and (*B. hydrogenotrophica*), which all grow on H_2_-CO_2_ as substrates to produce acetate as the main product. Homo-acetogens belonging to this family were also found to be present in kangaroos and will be discussed further [27].

Homo-acetogens in Ostrich

It was found that homo-acetogens are one of the dominant microbes in the gut of ostriches and play an important role in the H_2_ utilization [70,71,72,73]. Three different pathways were found to play the role of H_2_ sinks in ostrich; methanogenesis, acetogenesis, and sulfate reduction [70].

In vitro studies were conducted on the compartmentalized hindgut consisting of ceca, beginning and end of the proximal colon for harvesting variable populations of methanogens and homo-acetogens as dominant H_2_ utilizer [70,72,73]. The hindgut of the ostriches was reported to produce 52–76% of the daily intake of metabolizable energy, which was higher than the value reported for ruminants (60–70%) [72,73]. The ceca and the beginning of the colon showed the dominance of homo-acetogens (responsible for 26% and 22% of the total acetate produced), while the populations of methanogens increased and homo-acetogen populations decreased (12% of the total acetate produced) towards the proximal part of the colon. Reduced bile salt concentrations in the hindgut might be one of the reasons behind the dominance of methanogens, along with nutrition, as it was observed that fasting had a negative effect on methanogens while homo-acetogens remained unaffected [74,75]. Other factors possibly impacting the competitiveness of homo-acetogens over methanogens include the presence of protozoa and pH [71,76,77].

As per studies conducted on ostriches by Swart et al. in 1993, ostriches from Australia and New Zealand did not emit a significant amount of methane, while the ones in South America and Africa were found to do so. However, the ostriches considered for the studies were much younger than the ones tested by Fievez et al. in 2001 [70]. This was similar to the observations in most of the ruminants, where methanogens start appearing and becoming dominant in the gastrointestinal tract after the animal gets older [78,79]. However, a detailed analysis needs to be done to investigate the diverse population of the homo-acetogens present in the ceca and colon of the ostrich.

Homo-acetogens in Termites

An extensive study has been done by Breznak et al. in 1986 on reductive acetogenesis of H_2_-CO_2_ in the gut of wood- and soil-feeding termites. Acetate formation from H_2_-CO_2_ was found to play a major role as a source of energy and contributed to one-third of the respiratory requirement for the termites [38].

It was observed that homo-acetogenesis in wood- and grass-feeding termites were dominated by Spirochetes of genus *Treponema*, while Firmicutes acetogens dominated amongst the soil-feeding termites. The rate of acetogenesis coupled with the acetogen to archaea ratio showed the dominance of homo-acetogens and was observed to be higher in wood-feeding termites compared to their soil-feeding counterparts [80,81].

Studies on the hindgut of *Reticulitermes flavipes* (*R. flavipes*), a wood-feeding termite [82], showed a high acetogenic population in the central region where H_2_ concentration was maximum due to its production by Protozoa present there [83]. The methanogenic population dominated the gut epithelium, and the reasons for this are still vague and need more data before comprehension.

Curiously, it was also observed that the central lumen area of the hindgut of the termite had a high concentration of H_2_ up to 50 mbar, which is higher than the concentration required by most homo-acetogens [82]. This increase in the H_2_, which in turn increased the homo-acetogenic population, was mainly due to interspecies H_2_ transfer from the H_2_- producing Protozoa and Spirochetes, which get attached to Protozoa [84].

Spirochetes were other members of the microbiota showing acetogenic activity, forming half the population of prokaryotes present in the gut of termites [38,85]. Two strains of Spirochetes belonging to *Treponema primitia* sp. Nov. (ZAS-1 and ZAS-2) were isolated from the hindgut of *Zootermopsis angusticollis*, both producing acetate from H_2_-CO_2_ [38,80,86]. However, these strains showed a doubling time of 24–48 h, which was much higher compared to hydrogenotrophic methanogens. In addition to this, these strains also required yeast components and a cofactor solution for their growth in situ.

*Treponema azotonutricium* sp. Nov. strain (ZAS-9) was another strict anaerobic strain of Spirochetes isolated for termite gut, but it is an H_2_ producer instead of an H_2_ consumer. Like the earlier mentioned strains, ZAS-9 also had a long doubling time of 35–47 h and was found to tolerate only a slight range of temperature shift with optimal growth at 30 °C and no growth at 37 °C.

The reason for the long doubling time in the case of Spirochetes has been attributed to their primary focus on a specific function in the environment more than the rate and efficiency [87]. However, since it produced H_2_ in high concentrations (5.20 mol/mol maltose) when grown on complex sugars, this characteristic of H_2_ production can be beneficial for increasing its concentrations in the gut/rumen of the ruminants so that the mixed consortia of homo-acetogens would be able to grow and outcompete methanogens.

Similar H_2_ distribution was observed in the highly compartmentalized gut of soil-feeding termites [1,88,89]. The maximum concentration of H_2_ was found in the anterior portion of the gut [90], while its levels were negligible in the posterior region (<100 Pa). These increased levels in the anterior portion were due to the inter-compartmental transfer of H_2_. The increased availability of H_2_ in the anterior region made homo-acetogens dominant in this portion while methanogens were dominant in the posterior regions. Additionally, the anterior portion was reported to have extreme alkaline conditions [91,92], possibly inhibiting methanogens while still being tolerable for some homo-acetogens. Overall, it can be concluded that homo-acetogens have the potential to outgrow methanogens at higher concentrations of H_2_. Therefore, strategies contributing to increasing the level of H_2_ inside the gut or rumen systems can help autotrophic acetogenesis to outcompete methanogenesis.

Overall, the H_2_ distribution in the gut of termites shows that reductive acetogenesis is able to overtake methanogenesis if the H_2_ concentration is high [93]. For this reason, the hindgut of the wood- and grass-eating termites seem to be a pool of robust homo-acetogens along with H_2_ producing Protozoa and the Spirochetes, which together make a team successfully competing with methanogens.

Homo-acetogens in Pigs

Homo-acetogens have also been found in pigs. Suspensions from the hindgut of pigs showed the presence of both methanogens as well as homo-acetogens, but still, the methanogens were found to be dominant. Homo-acetogens could only grow and perform reductive acetogenesis after BES inhibition of methanogens [72].

Homo-acetogens in Pandas

Homo-acetogens were found in excrement from pandas (*Ailuropoda melanoleuca*), thereby indicating a probability of their presence in their digestive system [73]. The members of the Firmicutes and Proteobacteria families were the major homo-acetogenic species found in their droppings along with an unidentified third phylum. Among members of Firmicutes, Clostridiaceae, and Streptococcaceae were the most common families, whereas Enterobacteriaceae was the most dominant Proteobacteria [74].

As with other animals previously mentioned, there were differences in the acetogenic population profile based on the developmental age of the animal, with the animal gaining a more diverse microbiota as its ages into adult life, with three genera (*Actinomyces*, *Microbacterium*, and *Aeromonas*) found only in adults. Further investigation and research will be required for screening for homo-acetogenic population in the gut of pandas [74].

#### 2.2.2. Foregut Fermenters

In these animals, the stomach pouch is compartmentalized into at least two parts, one consisting of a fermentation chamber and the other capable of excreting the gastric acids. This form of digestion allows the animals to be more efficient in digesting biomass with higher fiber content when compared to hindgut fermenters [94].

Foregut fermentation occurs in a wide range of mammalian herbivores, most notably ruminants, and macropods. The foregut system of ruminants and macropods, though functionally similar to each other, have very different morphologies. The macropod’s foregut has a more tubular form, resembling more of an equine hindgut colon than the structure of the rumen.

The foregut fermentation system of macropods can be compared to a plug-flow system, with the feed being chewed only once at the initial ingestion and transferred in discrete boluses to the foregut. While the foregut fermentation system of ruminants functions more like a continuous-flow stirred tank reactor, where its large sacculated structure receives a less fine-grained feed that is mixed and fermented continually, with frequent regurgitation and re-chewing, and having a size selectivity for allowing only small particles to pass through it [66,95].

Homo-acetogens in Kangaroos

The compartmentalized stomach chamber analogous to the rumen in macropods is known as sacciform. In eastern grey and red kangaroos, the sacciform showed great diversity of homo-acetogens with most isolates from the Firmicutes belonging to *Clostridium* genus and at least 9 unidentified species [28,41,42].

Along with the morphological differences of the sacciform and the rumen, it is thought that the expansion limitations of it compared to the rumen is a major reason for active selectivity towards acetogens through the production of immune components against methanogens [96]. Studies on the digestive system of the macropods using radiological markers also showed the presence of biofilms containing immune agents in the second haustral pouch [90,91,92]. During digestion of food, these biofilms detach from the mucosal lining and carry the antimicrobial peptides (AMPs), innate lymphoid cells (ILCs), mucin, and immunoglobulins (IGs) against methanogens along with them. When these biofilms containing the microbes performing fermentation of complex sugars and the immune secretions mix with the food components, along with the digestion of food components, destruction of the methanogenic archaea may also possibly occur [95,96,97,98,99].

The feeding habits of ruminants also differ from macropods, due to the different morphology of the sacciform, as the biomass is ground only once by the animal, while ruminants rely on rumination to grind the feed material into finer particles, giving the fiber material greater retention time compared to other foregut fermenters [95,97].

The absence of methanogens in the sacciform, in turn, lead to increased availability of H_2_ for the homo-acetogens. Therefore, the H_2_ produced in the forestomach by fermentation of sugars is channelized towards homo-acetogenesis, making them interesting candidates for the investigation of competent homo-acetogens [97].

The methane production in eastern grey kangaroo (*Macropus gigantus*) and eastern wallaroo (*Macropus robustus*) was reported to be in the range of 0.5% and 1.8% of total gas present at the forestomach. It was observed that in vitro fermentation using the foregut sample from one of the selected kangaroos showed complete dominance of homo-acetogens and no methane production throughout the duration of the fermentation. Inoculation with a sample from another kangaroo showed that after a brief period of acetogenic growth, methanogens started growing and were the principle H_2_ consuming microbes present in the system. However, it was only after the loss of homo-acetogens that the methanogens were able to grow in case 2 [22,41].

Hence, it can be stated that homo-acetogens can be dominant in the kangaroos (foregut samples) and, in their presence, methanogenesis is not predominant. Simultaneous PCR amplification studies with formyltetrahydrofolate synthetase (FTHFS) genes showed the presence of 161 clones, of which 50 clones obtained from 13 different kangaroos were not found to match with the known sequences [22]. However, these were clustered in families of known homo-acetogens and they mainly showed sequence similarity to: *Sporomusa ovata*, *Sporomusa termitida*, *M. thermoacetica*, *C. magnum*, *Clostridium formicaceticum*, *A. woodii*, *Acetobacterium psammolithicum*, *Eubacterium limosum* (*E. limosum*), *Thermoanaerobacter kivui*, *Ruminococcus productus* (*R. productus*), and *Treponema* sp ZAS2 [22].

When reactor cultures inoculated with rumen fluid from cattle were augmented with acetogens from kangaroo forestomach cultures, it was found that the homo-acetogens were not able to sustain. The only acetogen that could sustain and maintain its population density was *Eubacterium cellulosolvens* (*E. cellulosolvens*) YE257 [22]. However, they were only able to grow after inhibition of methanogens with 2-bromoethanesulfonate (BES). It was later found that the acetogen *E. cellulosolvens* YE257 was present in all the kangaroos, and thus it seems to be robust, less fastidious, and host specific in nature. In the foregut samples from eastern grey and red kangaroos that were tested, along with acetogens, a group of Enterobacteria (*Escherichia coli* and *Shigella*) were also found to fix H_2_ for producing acetate. Three acetogen isolates, namely *E. cellulosolvens* YE257 (both kangaroos), YE266, and YE273 (only grey kangaroos) were found to be present in higher densities in the foregut of kangaroos vigorously consuming H_2_. All the three homo-acetogens showed distant relation (91% 16S rRNA similarity) to *Ruminococcus gnavus* [41].

Research on kangaroo forestomach and bovine rumen contents showed significant differences between kangaroos and cattle in the pathways of CO_2_ metabolism. In samples from bovine rumen fermentation, both grain-fed and grass-fed, methane was detectable in the headspace 3 h after inoculation and it continued to accumulate for the following 7 days of the experiment. In contrast, kangaroo culture fermentations produced a measurable quantity of methane only after 7 days of incubation, and this quantity was significantly (*p* < 0.01) lower than that in the rumen fermentations [100]. When the CO_2_ and bicarbonate were labeled to detect their usage by methanogenic and acetogenic populations in the rumen and kangaroo foregut, it was observed that maximal labeled population of ‘C’ was found to be incorporated in acetate in the case of kangaroos while the methane was produced only in small quantity and did not show any labeled C. From this, it can be construed that acetogenesis was a preferred pathway for the conversion of H_2_ and CO_2_ produced during digestion.

The most dominant species found by Godwin et al. in 2014 [42] in grey kangaroo samples shared maximum similarity to *Blautia coccoides*. Other closely related homo-acetogens were genera *Prevotella (Prevotella salivae* str. EPSA11; JCM 12084), *Streptococcus* (*S.* sp. YE54 (kangaroo forestomach), *Streptococcus mitis* str. MG3, *Streptococcus caballi* str. 151), *Oscillibacter* (*O. valericigenes* Sjm18-20), and *Lachnospiraceae*. Sequence analysis of the foregut samples from *Micropus giganteus*, an eastern grey kangaroo, using 16S rRNA genes, showed maximum bacterial population belonging to phylum Firmicutes with a majority belonging to order Clostridiales and Lactobacillales [28]. Homo-acetogens belonging to *Blautia* sps. and *Lachnospiraceae* sps. were also found to be present in tammar wallaby.

In situ fermentations of foregut samples performed at 39 °C (higher than its ideal temperature of 36–37 °C), ideal for rumen fermentation had no impact on the dominance of acetogenesis over methanogenesis. This indicates the robustness of the homo-acetogens in the foregut of the kangaroos [42]. Therefore, the higher temperature present at the rumen alone does not enhance the presence of homo-acetogens in rumen systems. Other factors such as anatomic differences of the forestomach might be playing an important role in the prevalence of methanogens in the rumen [22].

Homo-acetogens in Tammar wallaby

Tammar wallaby (*Macropus eugenii*) was also found to have reductive acetogenesis pathway dominating over methanogenesis. Methane emission from tammar wallaby was found to be 1–2% of the digestible energy intake, which was much lower than ruminants, which constituted ca. 10% of the digestible energy [96,101]. Overall, methane emission of 5.4 and 9.8% was observed in the red-necked wallaby (*Macropus rufogriseus*) and swamp wallaby (*Wallabia bicolor*) [41]. The comparative studies on the rumen systems and the forestomach of tammar wallaby showed the presence of novel homo-acetogens, which showed modifications in the gene sequences of the acetyl-CoA synthase (ACS) gene. Most of the homo-acetogens detected in both systems which were tested were found to be affiliated to Lachnospiraceae, while some homo-acetogens in the rumen belong to Ruminococcaeae/*Blautia* group. Homo-acetogens showing 99% similarity to *Blautia* sps. *Blautia hydrogenotrophica* were also found in the tammar wallaby [44]. Some sequences in the rumen system showed distant similarity with the Eubacteriaceae homo-acetogens and Deltaproteobacteria, but not with the samples from wallaby. A new mixotrophic strain designated as TWA4 was isolated from tammar wallaby. However, it required higher concentrations of H_2_ (>5 mM) for performing reductive acetogenesis—higher than *Methanobrevibacter smithii*, a dominant methanogen in the rumen [102]. Further studies are required on the forestomach of marsupials such as kangaroo and tammar wallaby to explore unique acetogenic communities capable of outcompeting methanogens [39].

Homo-acetogens in rumen

Homo-acetogens have the ability to grow on a variety of substrates other than H_2_-CO_2_, and hence are still a major population in the rumen. *Acetitomaculum ruminis* (*A. ruminis*) is one of the most conspicuous homo-acetogens residing in the rumen. *A. ruminis* was isolated from bovine rumen by culturing on 10% clarified rumen culture. It was found to produce ca. 2–8-fold higher amount of acetate using H_2_-CO_2_ (80:20) compared to N_2_-CO_2_ (80:20) headspace atmosphere [24].

*E. limosum* and its strains RF and S were isolated from the rumen of a young calf that was fed a molasses-based diet [103]. It was observed that *E. limosum* played an active role in maintaining low concentrations of H_2_, facilitating interspecies H_2_ transfer. Kinetic studies of the H_2_ uptake rate of *E. limosum* showed that it has a Ks value of 0.34 mM, which was much higher than 2.5–12 µM for methanogens, making them unsuitable candidates for methane mitigation [40,104].

To examine the diversity of the acetogenic bacteria, Rieu-Lesme et al. in 1998, isolated 13 acetogenic bacteria from rumen samples. The rumen samples included two young suckling lambs, two llamas, and two bison that were fed a grass-based diet. The isolates were all found to grow autotrophically on H_2_-CO_2_ and acetate was the only product that was detected. However, unlike methanogens, yeast extract was found to be pre-requisite for most of these strains. Future studies would be required for checking the competence of these homo-acetogens compared to methanogens found in the rumen [43].

Five strains of acetogenic bacteria were isolated from 3-, 1-, and 4-day-old newborns, suckling lambs that did not harbor any methanogens. The five strains—AA1, A94, A95, AA2, and A90—were found to only grow under autotrophic conditions while producing acetate as the sole end-product from a gas mixture of H_2_-CO_2_. The growth on H_2_-CO_2_ was enhanced by increasing the concentration of yeast extract. Isolated strain AA1 amongst the five strains exhibited a close phylogenetic relationship with *Clostridium difficile* (*C. difficile*), which has been isolated from various ecosystems such as soils, human gastrointestinal tracts, and animals all over the world, and, for the first time, in the rumen. However, unlike *C. difficile*, this strain showed the ability to grow on H_2_-CO_2_, but further investigation is needed for understanding its capability of acetate production using H_2_-CO_2_ [31].

Similarly, from the rumen of 1- and 3-year-old suckling lambs, two other strains of acetogenic bacteria B and Bie 41 were isolated with H_2_-CO_2_ as carbon and energy source. These strains were found to belong to *Clostridium* cluster XIVa and assigned as belonging to a new species: *Ruminococcus chinkii* sp. Nov. after 16S rRNA studies. Even though both strains converted H_2_-CO_2_ to acetic acid, no studies were done to compare these strains towards methanogens [43].

Six different species of acetogenic bacteria—*A. ruminis*, *E. limosum* strains ATCC 10825, and ATCC8486, *R. productus* ATCC 35244, Ser 5, and Ser 8—were isolated from 20 h old lambs, to assess their potential in outgrowing methanogens. Only *E. limosum* and Ser 5 strains were able to reduce methane production by 5% with the same amount of inoculum [25]. In conclusion, despite the addition of large concentrations of acetogenic bacteria, they were unable to compete for H_2_ with methanogens under normal circumstances.

While in this abovementioned context, the other bacteria were detected to have no effect on the production of methane. Both bacteria *E. limosum* strains ATCC8486 and Ser 5 exhibited a significant increase in the production of acetate and a decrease in H_2_ concentration when they were added to mixed ruminal microorganisms in the presence of methanogen inhibitor BES [105]. Overall, it was observed that the acetogens found in the rumen were not very competent competitors against hydrogenotrophic methanogens. It was only in the infant stages that the ruminants had significant populations of acetogens, being the main micro-organisms responsible for the turn-over of H_2_ in the rumen. However, once the animal matures, methanogens start to be more predominant in the rumen, leaving homo-acetogenesis playing a diminishing role for H_2_ metabolism in the rumen, until eventually taking over as the main hydrogen sink.

## 3. Rumen as a Bioprocess Reactor

The microbial production of acetic acid and other volatile fatty acids (VFA) in the rumen have a profound economic impact, as the breakdown of feed material by bacterial fermentation in the rumen is the main source of nutrient and energy of ruminants, and therefore its performance and efficiency affects the productivity of products derived from livestock such as milk, meat, wool, and leather [106,107]. The concentrations of different types of VFA in the rumen are directly correlated with the feed efficiency [108].

The rumen, therefore, has been described as a naturally occurring Consolidated Bioprocessing (CBP) reactor for the production of VFA and protein. In it, substrate hydrolysis and fermentation occur in the same reaction chamber [109]. Considering that there are over 3 billion ruminant livestock worldwide, the rumen could be viewed as the most used type of CBP process for the production of bioproducts [109,110]. Its understanding and characterization could allow for the development of robust CBP industrial processes, and it had been implemented as such in vitro to produce bioproducts, including the acetic acid [111,112,113,114].

The VFA production in the rumen, however, mostly occur through the oxidative breakdown of biomass, and not through the reduction of carbon dioxide. The archaea genus *Methanobrevibacter* are the predominant group of microbes responsible for reducing carbon dioxide and converting the majority of hydrogen, produced during fermentation, into methane [115,116,117,118]. Methanogens are an important part of the animal metabolism as they serve as a major hydrogen sink, recycling energy, and preventing feedback inhibition of rumen fermentative microbes, thus helping the animals to maintain the production of VFAs through the glycosylic pathway [119,120,121].

If there are no microbes in the rumen capable of performing the crucial role of hydrogen sink, the partial pressure of hydrogen increases. Hydrogen is produced by hydrogenase enzymes to oxidize ferredoxin by either phosphoroclastic reaction or redox couple reaction. When hydrogen accumulation occurs in the rumen, ferredoxin hydrogenases start to reduce ferredoxin, and acetate production from pyruvate by pyruvate ferredoxin oxidoreductase is hindered, consequently inhibiting the main pathway by which acetate is produced in the rumen [122,123].

As mentioned, homo-acetogens are a concurrent biological process of methanogenesis, competing for the same substrates but producing different products. The identified bacterial community in the rumen is much more diverse compared to methanogens, and they do not belong to a specific single taxonomic group, having diverse characteristics and sharing the capability for reductive acetogenic activity, though none of them have been identified to be obligated hydrogenotrophic.

Homo-acetogens can potentially substitute entirely the role of methanogens in the rumen; therefore, there has been a great effort on the implementation of biotechnological tools to understand and manipulate the ruminal microbiota towards the acetogenic pathway. The main types of biotechnological-based management for improving sustainability are focused on 5 categories: (1) Diet modification; (2) Feed transformation before consumption; (3) Defaunation; (4) Direct-fed microbes (DFM); (5) Chemical-based feed additives. These strategies aim at manipulating the fermentation of the rumen and improve the productivity of dairy and meat production through a decrease of energy loss through methane production [124].

Diet modification

It has been observed that the type of diet that the animal is fed changes the VFA produced by the animal. High-quality forage contains higher amounts of carbohydrates and starch compared to low-quality forage, altering the structure and functionality of microbial communities in the rumen and increasing the abundance of the Proteobacteria phylum and the family of Succinivibrionaceae, both capable of utilizing hydrogen to produce succinate, a precursor to propionate production. This might be explaining the lower methane production during high-quality forage feeding [125,126].

Feed transformation

When agricultural wastes such as corn stover, wheat, and rice straw are implemented in the ruminant’s diet, their high fiber contents tend to promote greater methane production by the animal. Feed transformation may involve biological, physical, and chemical treatments to improve digestibility in order to increase its nutritional value and lower its fiber content [127].

The feed particle size is an example of physical feed transformation which influences the motility of reticulo-rumen contractions, primarily responsible for mixing the digesta with the gas phase, and increasing the amount of dissolved gas, thus possibly altering the amount of substrates available for autotrophic microbes [128].

The application of chemical treatments to the biomass has also been studied, such as the use of Ammonia Fiber Expansion (AFEX) for increasing carbohydrate and protein of cereal straws and corn stover to improve its availability for rumen microorganisms [129].

Defaunation

Defaunation techniques are focused on the removal of Protozoa in the rumen. Protozoa constitute a large portion of the microbial biomass present at the rumen [130]. It is estimated that Protozoa contribute up to 37% of the total methane production in the rumen, due to its symbiotic relation to methanogens, serving as a host to protect methanogens from oxygen toxicity, and by providing H_2_ through interspecies H_2_ transfer [130]. They are, nevertheless, non-essential for the survival of the ruminant, and its defaunation with detergents has led to observable in vivo decrease of methane emissions of up to 49% [131].

Direct-Fed Microbes (DFMs)

DFMs are another management technique for methane mitigation. *Lactobacilli* spp. has been the main microbe used as DFMs. DFMs have been mainly used for helping ruminants to establish and maintain a healthy gut microbiota and prevent digestive disorders such as acidosis from high-quality forage feeding [132,133,134].

The implementation of a reductive acetogen as DFMs in the rumen has been attempted, but suppression of methanogens when achieved was only temporary, even in an association of other DFMs [25,135]. The utilization of homo-acetogens as DFMs has been performed only at experimental levels, with reports of a temporary decrease of 80% in the methane production [135].

Chemical-based feed additives

Common chemical feed additives used in the livestock industry, such as ionophores, have shown to have mitigation effects on rumen methane production, although this decrease is mostly associated with its antibiotic effects on Protozoa and bacteria, rather than by directly affecting the methanogen activity [136,137].

A new class of chemicals is being researched that would more specifically inhibit methanogens’ activity, usually targeting methyl-coenzyme M reductase (MCR), the enzyme directly responsible for the release of methane, as seen in Figure 2. One prominent direct inhibitor of MCR is 3-Nitrooxypropanol (3-NOP) which promoted up to 30% methane decrease in dairy cattle with no apparent side effects to the animal [138].

There are also MCR analogs, such as BES, that are able to effectively stop methane production in the rumen culture [112]. Inhibition from BES was observed in vivo for 4 days until tolerance to the chemical by methanogens was acquired, either by microbes present at the rumen being able to break down BES or by methanogens developing resistance to it [139]. This temporary inhibition observed in the rumen indicates that constitutive homo-acetogens found in the rumen are not suitable to take over the role as principal hydrogen sink, elucidating the importance for the identification of non-rumen homo-acetogens that are more competitive to methanogenesis [140].

## 4. Competition with Hydrogenotrophic Methanogenesis

Methanogenesis has been shown to dominate in rumen systems even in the presence of homo-acetogens. However, as discussed previously, there are certain species like kangaroos, wallabies, ostriches, rabbits, and termites where methanogenesis has been found to be suppressed [101,141,142].

Furthermore, it has been reported that in the absence of methanogenesis, there is significant production of acetate from H_2_-dependent CO_2_ reduction. The free energy change in H_2_-CO_2_ acetogenesis (−105 kJ per mol) however, is less efficient compared to methanogenesis from H_2_-CO_2_ (−136 kJ per mol) [12,143]. In spite of that, decreasing the pH to acidic conditions may influence the outcome of competition against methanogenesis [144].

Methanogenesis was first noticed by Gunsalus and Wolfe and called the Wolfe Cycle. Hydrogenotrophic production of methane, through the Wolfe Cycle, involves a cyclical pathway where one molecule of CO_2_ is converted to formyl-methanofuran (formyl-MFR) using ferredoxin, which is reduced using a H_2_ molecule by electron bifurcation (Figure 1). Formyl-MFR is then converted to methyl-tetrahydromethanopterin complex through a series of enzymatically-driven reactions involving three more molecules of H_2_. This complex combines with coenzymes M and B resulting in the release of methane. This coenzyme-complex formed is later recycled into the system using one molecule of H_2_ by electron bifurcation. The entire process requires four molecules of H_2_ and one molecule of CO_2_ generating a Gibbs free energy (ΔG0′) of −136 kJ/moles [145].

The ability of methanogens to grow in diverse habitats, with minimal nutrient requirements makes them able to survive in extreme conditions. In most of these anoxic environments, H_2_ is produced in low concentrations as an intermediate during anaerobic fermentation of organic compounds. Hydrogen plays an important role as an electron donor for the methanogen and acetogenic bacteria present in the anoxic habitat [145,146,147,148]. However, since methanogens have lower K_s_ values than most of the homo-acetogens, even when the homo-acetogens have higher V_max_, methanogens are able to outcompete homo-acetogens at a limited H_2_ concentration [145,149,150,151].

The H_2_ concentration also plays another significant role—its decrease below the normal concentration increases the Gibbs free energy to a point where an exergonic reaction is not possible to facilitate H_2_ uptake. The concentration of H_2_ in the rumen is lower than the threshold required for homo-acetogens, but enough for the uptake by the methanogens. Therefore, methanogens outcompete homo-acetogens even though both have the ability to metabolize H_2_-CO_2_ [59,60,145,152].

Most of the H_2_ threshold studies related to homo-acetogens and methanogens have been performed in situ using externally supplied H_2_. However, there are various other mechanisms, which involve the direct transfer of electrons between the donor microbe and the receptor microbe through interspecies H_2_ transfer. The close vicinity between the H_2_-producing microbes and the H_2_-utilizing microbes affected the local concentrations of H_2_; thereby changing the energetics of the process compared to a direct transfer in the liquid phase [35,56,57,58].

Therefore, even though these anoxic environments generally are inhabited with both hydrogenotrophic methanogens and homo-acetogens, the possibilities for homo-acetogens to prevail over methanogens by using interspecies hydrogen transfer still needs to be investigated. Besides, homo-acetogens can grow heterotrophically by utilizing sugars and other substrates present in the anaerobic environments, without having to rely on higher H_2_ concentrations [153].

## 5. Discussion

In this review, we have investigated various homo-acetogens found in different habitats to study their growth characteristics and competitive relationship with hydrogenotrophic methanogens. Their ability to grow on gaseous substrates such as H_2_/CO_2_/CO makes them interesting candidates as potential H_2_ scavengers in place of methanogens.

Methanogens are robust archaea, which can grow in different environments with minimal nutritional requirements. They are able to grow hydrogenotrophically at hydrogen concentrations of 0.02–0.1 mbar, and thus can survive on a minimum hydrogen threshold of 0.01–0.04 mbar existing in anaerobic environments like sediments and digestive systems of ruminants. Most of the homo-acetogens found in the anaerobic sediments also had the ability to grow hydrogenotrophically on H_2_. However, the concentration of H_2_ in many natural environments and animal gut systems is too low for allowing the growth of the homo-acetogens.

Unlike ruminants, marsupials such as kangaroos, tammar wallabies; birds like ostrich, and wood- and grass-feeding termites have predominance of acetogenesis over methanogenesis. Investigation into different parts of the digestion system of these animals showed that one of the major factors governing the dominance of homo-acetogens was the spatial distribution of H_2_ inside their digestive system. There were specific regions in their digestive system with higher concentrations of H_2_ which favored the growth of homo-acetogens. For example, in the case of marsupials, like kangaroos and tammar wallabies, the concentration of H_2_ was higher in the forestomach. In the case of termites, H_2_ was high in the central region of the hindgut. Methanogens were also found to coexist in the regions of lower H_2_ levels.

An anatomical study conducted by Leng in 2018 showed that, in the case of marsupials, the sensitivity of their digestive system towards higher concentrations of gases may have prevented the formation of methane by inhibiting methanogens and facilitating homo-acetogens by excreting immunogenic components, which acted against the methanogens and made H_2_ available for homo-acetogenesis.

Secondly, the microbial community, type, and subspecies of homo-acetogens found in ruminants were also found to differ slightly from the environments dominated by homo-acetogenesis. Most of the types of homo-acetogens found in different habitats showed sequence homogeneity to the commonly found species and were similar to each other. However, each of the habitats had its own set of novel homo-acetogens, which, besides having their own unique gene sub-sets, shared sequences with most of the commonly found species. For example, an FTHFS clone library from isolates of different macropods showed that of the 161 clones obtained, 50 clones did not match any known library sequences. This indicates the presence of novel homo-acetogens and the possibility of strains possessing more competitive H_2_-utilizing kinetics for successful competition with hydrogenotrophic methanogens. When the homo-acetogens present in the rumen and marsupials were compared, marsupials like kangaroo and tammar wallaby showed the presence of different acetogenic species like *A. woodii*, *C. magnum*, *Clostridium formicaceticum* (AF295075), *S. ovata*, *S. termitida*, *M. thermoacetica*, along with *Blautia* species such as *B. coccoides*, *B. hydrogenotrophica*, and *Ruminococcaceae* species including *R. productus* ATCC 35244, *Streptococcus* species., *Thermoanerobacter* AF29, *Treponema species*., *Prevorella salvae,* and a few more (Table 1), were not observed to be very prevalent in the rumen systems. The acetogenic species present in the acetogenesis-dominant environments were from genera *Sporomusa*, *Blautia*, *Clostridium*, *Treponema*, *Oscillibacter*, *Prevotella*, *Thermoanaerobacter*, and *Acetobacter*. Hence, there are many strains of homo-acetogens, which have not currently been tested for their capabilities for competing with hydrogenotrophic methanogens.

From the reviewed literature, environments with a dominance of homo-acetogens are not generally limited to a single species. Instead, many homo-acetogenic bacteria seem to be involved. There are various factors such as an increased H_2_ concentration due to the presence of specific H_2_-producing populations, that might work synergistically with homo-acetogens. Interspecies H_2_ transfer might further be an important factor besides the occurrence of bile acid/gastric juices in the stomach, which can affect pH in the system. Anatomy of the digestive system along with specific immunological factors have further been described as important factors, possibly changing the competition between methanogens and homo-acetogens [97]. Therefore, a deeper understanding of the dominance of homo-acetogens in marsupials in comparison to ruminants is needed.

## 6. Conclusions

In this review, we examined different environments for potential bioprospection of homo-acetogens. However, in these environments, the homo-acetogens used their diverse metabolic flexibility towards organic substrates such as sugars. Furthermore, these environments often have steady-state H_2_ concentrations lower than the threshold concentration required for homo-acetogenic growth.

Yet, studies on kangaroos, wallabies, ostrich, and termites showed that homo-acetogens were dominant in these species, and thus, under the right conditions, they can be potential candidates for having homo-acetogens capable to outcompete methanogens in the ruminants.

The diversity of homo-acetogens found in the digestive tract of macropods elucidates that overcoming hydrogenotrophic methanogenesis may require a cumulative action of different types of homo-acetogens working symbiotically with one another and with other non-acetogenic bacteria. Understanding the mechanisms behind these relationships of the microbes might pave the way for solutions whereby hydrogenotrophic methanogenesis can be substituted with acetogenesis in ruminants in the future.

## Figures and Tables

**Figure 1 microorganisms-10-00397-f001:**
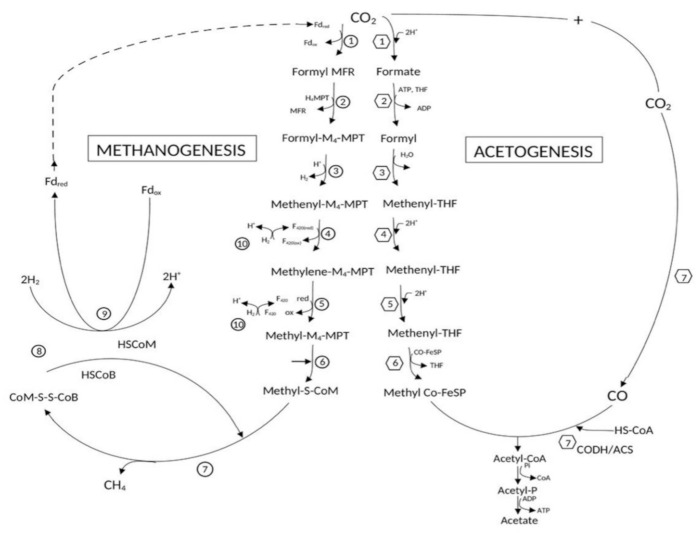
Hydrogenotrophic growth of methanogens using Wolfe cycle/pathway and homo-acetogens using WL pathway for fixation CO_2_ and H_2_. The circles numbered from 1–10 represent the enzymes involved in fixation of H_2_-CO_2_ in Wolfe pathway for methanogenesis; formylmethanofuran dehydrogenase, 2—formylmethanofuran/H_4_MPT reductase, 3—methenyl-H_4_MPT cyclohydrolase, 4—methylene-H_4_PT dehydrogenase, 5—methylene-H_4_MPT reductase, 6—methyl-H_4_MPT/coenzyme M methyltransferase, 7—methyl-coenzyme M reductase, 8—electron-bifurcating hydrogenase-heterodisulfide reductase complex, 9—F_420_-reducing hydrogenase, 10—energy-converting hydrogenase (catalyzes sodium motive force-driven reduction of ferredoxin by H_2_). The hexagons numbered from 1–7 represent enzymes involved in the fixation of H_2_-CO_2_ in WL pathway; 1—formyl dehydrogenase, 2—formyl tetrahydrofolate synthase, 3—formyl tetrahydrofolate cyclohydrolase, 4—methylene tetrahydrofolate dehydrogenase, 5—ethylene tetrahydrofolate reductase, 6—methyl transferase, 7—carbon monoxide dehydrogenase/acetyl-CoA synthase.

**Figure 2 microorganisms-10-00397-f002:**
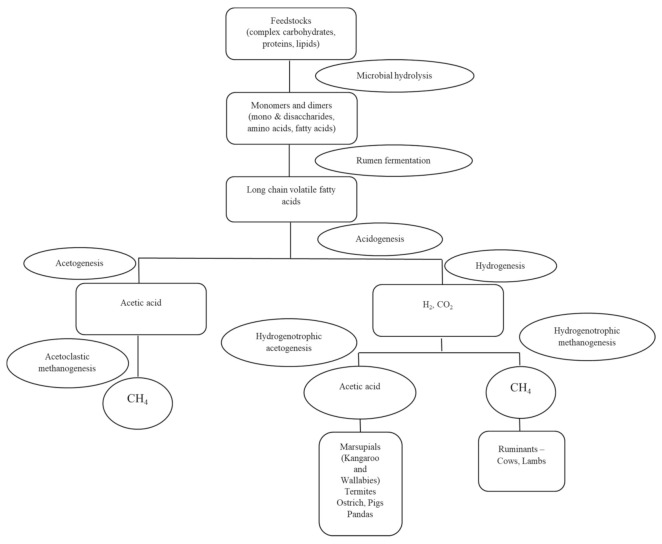
Summary of the metabolic scheme of organic matter degradation through fermentation that occurs in the digestive tract of animals, leading to the formation of methane and acetic acid.

**Table 1 microorganisms-10-00397-t001:** Homo-acetogens isolated from different habitats. (O) represents optimal temperature.

Micro-Organism	Habitat	Growth Temperature	Growth pH	References
*Acetobacterium woodii*	Sediments	30 °C	6.8–7	ATCC; [12]
*Acetobacterium woodii* (AF29570)	Kangaroo	30 °C	7	[12,22]
*Acetobacterium carbinolicum*	Freshwater sediments	30 °C	–	ATCC; DSMZ; [12]
*Acetobacterium malicum*	Sediments	30 °C	–	ATCC; [12]
*Acetobacterium* HP4	Lake sediments	1–25 °C psychrotrophic	–	[23]
*Acetobacterium ruminis* ATCC 10825	20 h old lamb, rumen fluid	30 °C	7.2	ATCC; [24,25]
*Acetobacterium psammolithicum*	Subsurface sandstone, kangaroo	30 °C	6.8	[22,26]
*Clostradiaceae* sps.	Kangaroo, rabbit	–	–	[27,28]
*Clostridium drakei*	Coal mine sediments	30–37 °C	5.4–7.5	DSMZ
*Clostridium glycolicum* RD-1	Sea grass roots	22–37 °C	7.4–7.6	[29]
*Clostridium carboxidivoran P7*	Sediments	35–38 °C	6.2	DSMZ; [12]
*Clostridium scatologenes*	Coal mine pond sediments, soil	37 °C	6.3	[30]
*Clostridium difficile*	Rumen, sediments	37 °C	–	[31,32]
*Clostridium magnum*	Kangaroo, freshwater sediment	30 °C	–	[12,22,33]
*Clostridium formicaceticum* (AF295075)	Kangaroo	37 °	7.6	[22,33]
*Sporomusa malonica*	Fresh water sediments	30 °C	–	DSMZ; [12,34]
*Sporomusa paucivorans*	Lake sediments	34 °C	6.7	[35]
*Sporomusa* DR6	Rice field soil	–	–	[36]
*Sporomusa* DR1/8	Rice field soil	–	–	[36]
*Sporomusa ovata* (AF295708)	Kangaroo	30 °C	–	DSMZ; [22]
*Sporomusa termitida* (AF295075)	Kangaroo	30 °C	7.2	[37]
*Sporomusa termitida*	Gut of wood-eating termite	30 °C	7.2	[37]
*Morella thermoacetica* (J02911)	Kangaroo	55–60 °C	5.7–7.7, 6.8	[22]
*Eubacteriaceae*	Tammar wallaby	35–40 °C, 37 °C ideal	–	[38]
*Eubacterium limosum*	Rumen fluid, sheep	30–45 °C, 37 °C (O)	5–7.5, 7 (O)	[25,39]
*Eubacterium limosum* ATCC8486	Kangaroo, 20 h old lamb	37 °C	7.3 ± 0.2	ATCC; [22]
*Eubacterium limosum* (ELI494825)	Kangaroo	30–45 °C, 37 °C (O)	5–7.5, 7 (O)	[22,39]
*Ruminococcaceae*	Kangaroo, rabbit	–	–	[27,40]
*Ruminococcus Productus* ATCC 35244	Rumen, kangaroo	37 °C	7	ATCC; [25,40]
*Ruminococcus chinkii* sp.	rumen	–	–	[41]
*Blautia* sp.	Kangaroo, tammar wallaby, rumen, rabbit	37 °C	7 ± 0.2	ATCC; [27,39,42]
*Blautia coccoides*	Kangaroo, rabbit	37 °C	7	ATCC; [27,43]
*Blautia hydrogenotrophica*	Tammar wallaby, rabbit	35–37 °C	7 ± 0.2	ATCC; [27,42]
*Blautia schinkii*	rabbit	–	*–*	[27]
*Prevotella salivae* EPSA11	Kangaroo	37 °C	7–7.3	DSMZ; [27]
*Prevotella salivae* JCM 12084	Kangaroo	37 °C	7.6–7.8	JCM; [43]
*Streptococcus* YE54	Kangaroo	*–*	*–*	[43]
*Streptococcus mitis* MG3	Kangaroo	–	–	[43]
*Streptococcus caballi* 151	Kangaroo	35 °C	7.3–7.4 ± 0.2	ATCC; [43]
*Oscillibacter valericigenes* Sjm18–20	Kangaroo	15–35 °C, 30 °C (O)	–	[44]
*Treponema* sp. ZAS1	Termites	22–32 °C, 30 (O)	–	[22,45]
*Treponema* sp. ZAS2 *(TSP494823)*	Kangaroo, termites	22–32 °C, 30 °C (O)	–	[22,45]
*Treponema azotonutricium* ZAS9	termites	30 °C	–	[45]
*Thermoanaerobacter kivui* (AF29)	Kangaroo	50–72 °C, 60 °C (O)	5.3–7.3	DSMZ; [22]
YE257	Grey and red kangaroo	*–*	*–*	[22]
YE266	Grey kangaroo	*–*	*–*	[40]
YE273	Grey kangaroo	*–*	*–*	[40]
Ser 5	20 h old lamb	*–*	*–*	[25]
Ser 8	20 h old lamb	*–*	*–*	[25]
*Lachnospiraceae* sps.	Kangaroo, tammar wallaby, rabbit	–	–	[27,42,44]

## Data Availability

Not Applicable.

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
