# Peer review of "Homo-Acetogens: Their Metabolism and Competitive Relationship with Hydrogenotrophic Methanogens"

_microorganisms, 2022, doi:10.3390/microorganisms10020397_

Round 1
Reviewer 1 Report
Karekar et al. present a relevant review that addresses many aspects related to homo-acetogens. The topic is especially relevant considering the potential utilization of this group of microorganisms in circular economy and they might also play a role in methane emission mitigation from ruminants.
I have only few remarks:
The review contains some repetitive information, try to reduce them. (e.g.Line 205-215, Line 483-485, Line 516-525 (the two paragraphs about starch can be merged)etc.)
A journal style description remained in the text (Line 96-108). Please, eliminate it.
I would rename section 4 to “Competition with hydrogenotrophic methanogenesis”
It would be important to include the topic about the role of homo-acetogens in biomethanation processes when external H2 is injected (biological biogas upgrade or in situ and ex situ biomethanation). It would fit to section 4.
The names of genera and species are not always in italics. Please, proof read your text again.
Few examples: Line 286, Line 384
Figure 2. Include Syntrophic acetate oxidation in the figure. Please, also improve the quality of the figure (some texts are out of the text boxes in the pdf version).
Please also use the non-hyphenated writing style for in situ, ex situ, in vitro etc. These Latin terms are written often incorrectly in the literature.
Write homo-acetogen(esis) in a uniform way (check Line 145 and 500)
Line 184: Unit is missing (re-edit superscript)
Line 193. A word is missing from the beginning of the sentence.
Line 245: Use the term “microbiota” instead of “microflora”.
Line 339: % of what
Line 350-354: Do not provide the accession numbers (of the sequences) but write the full name of the microorganisms. The genus names are also important.
Line 355: ..grown with rumen content??
Line 385: Write 16S rRNA gene instead of 16S rDNA
Line 392-395: Rephrase this sentence. It is very confusing in its current form.
Line 448: 16S with capital S
Line 502: Do you mean “bacterial communities”? (bacteria population is surely a wrong term here)
Line 662-664: With isolation in microbiology strains or isolates are obtained and not clones. Please, check the original paper and rephrase the sentence accordingly.
Author Response
-The review contains some repetitive information, try to reduce them. (e.g.Line 205-215, Line 483-485, Line 516-525 (the two paragraphs about starch can be merged)etc.)
- 205-215 line revised (now line 196-204)
- Line 483-485 revised (now line 472-478)
- 516-525 lines revised (now line 502-509)
-A journal-style description remained in the text (Line 96-108). Please, eliminate it.
- Section deleted
-I would rename section 4 to “Competition with hydrogenotrophic methanogenesis”
- Section renamed to suggested title
-It would be important to include the topic about the role of homo-acetogens in biomethanation processes when external H2 is injected (biological biogas upgrade or in situ and ex situ biomethanation). It would fit to section 4.
- We feel that that the scope of this article is not to address bioprocesses for production of biomethane or biogas upgradation, but only to describe the acetogens, and the concurrent biological process that is methanogen as it is mostly found in nature.
-The names of genera and species are not always in italics. Please, proofread your text again.
- The names were revised and proofread
-Figure 2. Include Syntrophic acetate oxidation in the figure. Please, also improve the quality of the figure (some texts are out of the text boxes in the pdf version).
- I have not found literature describing Syntrophic Acetate Oxidation in digestive track of animals, only found literature pertaining to Anaerobic Digestors. Figure 2 intents to show the process that occurs inside hindgut and foregut fermenters. The description of Figure 2 was misleading and therefore revised to better describe the goal of the Figure to describe the fermentation scheme found in the GI tract of animals.
-Please also use the non-hyphenated writing style for in situ, ex situ, in vitro etc. These Latin terms are written often incorrectly in the literature.
- Latin terms were corrected for the correct format
-Write homo-acetogen(esis) in a uniform way (check Line 145 and 500)
- Homo-acetogens word was put into a single format of spelling
-Line 184: Unit is missing (re-edit superscript)
- Units for gene expression were added (now line 172)
-Line 193. A word is missing from the beginning of the sentence.
- The missing word “homo-acetogen” added (now line 183)
-Line 245: Use the term “microbiota” instead of “microflora”.
- The word was corrected (now line 231)
-Line 339: % of what
- The description of a percentage was added (now line 325)
-Line 350-354: Do not provide the accession numbers (of the sequences) but write the full name of the microorganisms. The genus names are also important.
- Name corrections made and accession number taken out (now line 340-343)
-Line 355: ..grown with rumen content??
- A better description of reactor content was made (now line 342)
-Line 385: Write 16S rRNA gene instead of 16S rDNA
- Corrected (now line 373)
-Line 392-395: Rephrase this sentence. It is very confusing in its current form.
- The sentence was rephrased ( now line 380-383)
-Line 448: 16S with capital S
- Capitalized ( now line 439)
-Line 502: Do you mean “bacterial communities”? (bacteria population is surely a wrong term here)
- Term changed to bacterial community ( now line 489)
-Line 662-664: With isolation in microbiology strains or isolates are obtained and not clones. Please, check the original paper and rephrase the sentence accordingly.
- Rephrased to refer clone reference from clone library, not isolates. (now line 645-649)
We appreciate for taking your time to review our article, your contribution surely helped us to better express the concepts and ideas that we presented in the article.
Reviewer 2 Report
The manuscript presents a large and complex study of homo acetogenic bacteria, and the possibility of using their properties to inhibit methanogenesis in the livestock industry. The competition of homo-acetogenic bacteria with the hydrogenotrophic methanogens in the ruminants can be directed according to the conditions and microbiota of the intestine. The authors propose to decrease the process of methanogenesis and increase that of acetogenesis in ruminants in the future by using these bacteria and manipulating the ruminal microbiota towards the acetogenic pathway.
Comments (in order of text)
- For figures and tables it should be specified which bibliographic reference was used, if there are no original contributions
- The writing of microorganisms should be checked and corrected throughout the text. The name of the species must be written in italics, while the order, family, etc. – non italic.
- Lines 42, 43 - there is a phrase that should be reformulated.
- 96 - 106 - it seems that these paragraphs are not part of this manuscript, they should be checked and deleted
- Table 1 “Clostridium carboxidivoran” - carboxidivorans ”
- Table 1 - “Ruminococcus Productus” Ruminococcus productus
- Table 1 - “Blautia sps.” - Blautia sp; Treponema sp - the abbreviation sp should be written in non-italic characters; “Thermoanerobacter kivui” –Thermoanaerobacter; "Lachnospiraceae sps." - sp
- In general, the way of writing the name of microorganisms contains quite a lot of mistakes, so it must be checked and corrected throughout the text.
- 126 “generas” - genera
- 129 “C. carboxidivoran ”- carboxidivorans
- 130 "Sporomusa" must be written in italics
- 193 - ??
- 252 - “sp” - sp non italic
- 325 - “the antimicrobial peptides (APMs)” - AMPs
- 348 - "FTHFS genes" - formyltetrahydrofolate synthetase (FTHFS)
- 351-353 - the name of the microorganism must be written in full, without abbreviation if it is used for the first time in the text
- 358 - "YE257" is the strain identification code, the name of the microorganism (species) should be written first, then the strain ID, ie: Eubacterium cellulosolvens YE257. The same should be corrected in the whole text where only the strain code is written.
- 368 CO2
- 384 - Micropus giganteus - italic
- 386 - the name of the order must be in non-italic characters
- 387 - Lactobacillus is a genus not an order
- 405, 406 - "to Lachnospiraceae while some homo-acetogens in rumen also belonged to Ruminococcaeae" - family name - non italic
- 409 - Eubacteriaceae - Non italic
- 420 - “It was isolated from bovine rumen by culturing the rumen fluid on H2-CO2.” – the expression should be checked because the reference states that it was grown on "minimal medium containing 10% clarified rumen fluid under either H2: CO2 (80:20) or N2: CO2 (80:20) headspace atmosphere".
- 422 - E. limosum - Eubacterium limosum
- 426 - uM?
- 473 - VFA explanation is only on line 491 -volatile fatty acids (VFA)
- 444, 445 - “Similarly, from the rumen of 1- and 3-year-old suckling lambs, two other strains of acetogenic bacteria B and Bie 41 were isolated on H2-CO2”. - were isolated from H2-CO2 culture media
- 579 – “tic acid his is a figure”. ??
- 593 MFR - formyl-MFR
- 595 methyl-H4MPT - what is H4MPT - Tetrahydromethanopterin
- 673-675- "The acetogenic species present in the acetogenesis dominant environments were Sporomusa, Lachnospiraceae, Blautia, Treponema, Oscillobacter, Prevotella, Thermoanaerobacter, Firmicutes, Ruminococcaceae, Clostridiaceae, Acetobacteria" - are listed different genera and families, these are not species
References - should be checked, some are incomplete, for example ref 86, 152. In addition, there are many references from the 90's and quite a few from the last 5 years.
Author Response
Comments (in order of text)
- For figures and tables it should be specified which bibliographic reference was used, if there are no original contributions
- References added to Table 1
- The writing of microorganisms should be checked and corrected throughout the text. The name of the species must be written in italics, while the order, family, etc. – non italic.
- Names were proofread
- Lines 42, 43 - there is a phrase that should be reformulated.
- Phrase removed, the same information is discussed later in the article ( line 57)
- 96 - 106 - it seems that these paragraphs are not part of this manuscript, they should be checked and deleted
- Paragraphs removed
- Table 1 “Clostridium carboxidivoran” - carboxidivorans ”
- Name was corrected
- Table 1 - “Ruminococcus Productus” Ruminococcus productus
- Name was corrected
- Table 1 - “Blautia sps.” - Blautia sp; Treponema sp - the abbreviation sp should be written in non-italic characters; “Thermoanerobacter kivui” –Thermoanaerobacter; "Lachnospiraceae sps." – sp
- Names corrected in Table 1
- In general, the way of writing the name of microorganisms contains quite a lot of mistakes, so it must be checked and corrected throughout the text.
- Names corrected
- 126 “generas” – genera
- Word corrected (now line 112)
- 129 “C. carboxidivoran ”- carboxidivorans
- Name corrected
- 130 "Sporomusa" must be written in italics
- Name corrected to italic writing
- 193 - ??
- Missing word added (Now line 183)
- 252 - “sp” - sp non italic
- Spelling corrected (Now line 238)
- 325 - “the antimicrobial peptides (APMs)” – AMPs
- Word correct (now line 311)
- 348 - "FTHFS genes" - formyltetrahydrofolate synthetase (FTHFS)
- Word corrected now line 335)
- 351-353 - the name of the microorganism must be written in full, without abbreviation if it is used for the first time in the text
- Complete names were written, except for the ones that have been mentioned in prior paragraphs ( now line 338)
- 358 - "YE257" is the strain identification code, the name of the microorganism (species) should be written first, then the strain ID, ie: Eubacterium cellulosolvens YE257. The same should be corrected in the whole text where only the strain code is written.
- Microbe’s name added throughout the text ( now line 345)
- 368 CO2
- Subscript added ( now line 356)
- 384 - Micropus giganteus – italic
- Name changed to italic format (now line 373)
- 386 - the name of the order must be in non-italic characters
- Names changed to non-italic format (now line 374)
- 387 - Lactobacillus is a genus, not an order
- Lactobacillales order was corrected ( now line 375)
- 405, 406 - "to Lachnospiraceae while some homo-acetogens in rumen also belonged to Ruminococcaeae" - family name - non italic
- Name changed to non-italic format (now line 393)
- 409 - Eubacteriaceae - Non italic
- Name change to non-italic format (now line 397)
- 420 - “It was isolated from bovine rumen by culturing the rumen fluid on H2-CO2.” – the expression should be checked because the reference states that it was grown on "minimal medium containing 10% clarified rumen fluid under either H2: CO2 (80:20) or N2: CO2 (80:20) headspace atmosphere".
- Description of higher production in reference to Nitrogen:Carbon Dioxide headspace atmosphere made, as well adding the description of the use of 10% clarified rumen culture. ( now line 407)
- 422 - E. limosum - Eubacterium limosum
- The abbreviation was maintained, with abbreviations introduced in prior paragraphs. (now line 411)
- 426 - uM?
- The correct micro symbol used (now line 415)
- 473 - VFA explanation is only on line 491 -volatile fatty acids (VFA)
- Correction made for VFA explanation corrected in line 458
- 444, 445 - “Similarly, from the rumen of 1- and 3-year-old suckling lambs, two other strains of acetogenic bacteria B and Bie 41 were isolated on H2-CO2”. - were isolated from H2-CO2 culture media
- Change word to “from” (now line 435)
- 579 – “tic acid his is a figure”. ??
- Description of Figure 2 was clarified and corrected (now line 563)
- 593 MFR - formyl-MFR
- The abbreviation was corrected (now line 576)
- 595 methyl-H4MPT - what is H4MPT - Tetrahydromethanopterin
- The description was corrected (now line 578)
- 673-675- "The acetogenic species present in the acetogenesis dominant environments were Sporomusa, Lachnospiraceae, Blautia, Treponema, Oscillobacter, Prevotella, Thermoanaerobacter, Firmicutes, Ruminococcaceae, Clostridiaceae, Acetobacteria" - are listed different genera and families, these are not species
- The list of microorganisms was changed to describe their respective genus ( now line 656)
References - should be checked, some are incomplete, for example ref 86, 152. In addition, there are many references from the 90's and quite a few from the last 5 years.
- References were formatted according to using Endnote MDPI template.
We appreciate your time in reviewing our article. Your comments and input helped us to improve it and they were much appreciated.